# Antibiotic Use in Broiler Poultry Farms in Kathmandu Valley of Nepal: Which Antibiotics and Why?

**DOI:** 10.3390/tropicalmed6020047

**Published:** 2021-04-05

**Authors:** Ananta Koirala, Priyanka Bhandari, Hemant Deepak Shewade, Wenjing Tao, Badri Thapa, Robert Terry, Rony Zachariah, Surendra Karki

**Affiliations:** 1Ministry of Agriculture and Livestock Development, Kathmandu 44600, Nepal; 2Institute of Agriculture and Animal Science, Post Graduate Campus, Tribhuvan University, Kirtipur 44618, Nepal; bhandaripriyanka24@gmail.com; 3International Union Against TB and Lung Disease (The Union), 75006 Paris, France; hshewade@theunion.org; 4International Union Against TB and Lung Disease (The Union), New Delhi 110016, India; 5Unit for Antibiotics and Infection Control, Public Health Agency of Sweden, SE-17182 Stockholm, Sweden; wenjing.tao@fohm.se; 6World Health Organization, Yangon 11201, Myanmar; thapab@who.int; 7Special Programme for Research and Training in Tropical Diseases (TDR), 1211 Geneva, Switzerland; terryr@who.int (R.T.); zachariahr@who.int (R.Z.); 8Food and Agriculture Organization of the UN (FAO), Emergency Center for Transboundary Animal Diseases (ECTAD), Kathmandu 44600, Nepal; surendra.karki@fao.org

**Keywords:** antimicrobial resistance, One Health, antibiotics stewardship, SORT IT, food producing animals, antibiotic use

## Abstract

Inappropriate antibiotic use in food-producing animals is associated with the emergence and spread of antibiotic resistance. In industrial broiler poultry farms in three districts of Kathmandu valley, Nepal, we assessed antibiotic use prevalence, and their classes, types, and quantities. A cross-sectional questionnaire study involving field visits to large poultry farms (flock size ≥ 3000) of the Kathmandu, Bhaktapur, and Lalitpur districts was conducted. Of 30 farms (total flock size 104,200; range 3000–6000), prevalence of antibiotic use was 90% (95% CI: 73–98%). Six (22%) farms used antibiotics as prophylaxis, while 21 (78%) used it for therapeutics. Seven antibiotics from six classes (including quinolones, macrolides, and polymyxins) were used. The most commonly used antibiotics were tylosin (47%), colistin (47%), and dual therapies with neomycin and doxycycline (33%). A total of 50,000 grams of antibiotics (total weight including active and inactive ingredients) were used (0.5 grams/chicken/45 days of flock life) with eight (26%) farms using more than two antibiotics. No farms had records on clinical indications for prophylaxis or treatment. No post-mortem records of sick birds were available. Prevalence of antibiotic use in broiler farms of Kathmandu valley is high and includes “highest priority critically important antibiotics” for human use, with direct implications on public health.

## 1. Introduction

The World Health Organizations’ (WHO) global action plan on antimicrobial resistance (AMR) emphasizes the “One Health” approach to tackle AMR [1]. This approach is one that englobes human, animal, food chain, environment, and the interconnections between them as one entity. Many of the same microbes infect animals and humans, as they share the eco-systems they live in. Efforts by just one sector cannot prevent or eliminate the problem. Drug-resistant microbes can be transmitted between animals and humans through direct contact between animals and humans or through contaminated food; so to effectively contain it, a well-coordinated approach in humans and in animals is required. The use of antibiotics in human medicine, veterinary medicine, and agriculture has been associated with the rise of antibiotic resistance [2]. Interventions that restrict or minimize antibiotic use in food-producing animals can reduce antibiotic resistant bacteria in these animals by up to 39% [2].

The use of antibiotics in animals started as early as the 1940s when it was discovered that feeding animals with an antibiotic (tetracycline) resulted in improved growth [3]. Today, approximately 80% of animals used in food production are given antibiotics at some point either as growth promoters, as prevention from infections in crowded and poor hygiene conditions (prophylactic use), or to treat infections (therapeutic use) [3,4].

Injudicious use of antibiotics, particularly those considered important and critical for use in humans, can result in AMR, such as in *Escherichia coli* [5,6]. Antibiotic resistant bacteria from animals including poultry may be transmitted to humans via direct contact with them and their products (meat or milk) during handling and slaughtering, consumption of contaminated food (farm-to-fork transmission), or environmental contamination through animal waste [7,8,9]. Ingesting such antibiotic resistant bacteria can then generate resistance in humans through lateral transfer of resistance (through plasmids) to human *E. coli* flora sharing the same ecological niche [8,10]. In South East Asia, up to 80% of animal waste is untreated and contaminates ground water, surface water, soil, and crops [9]. The majority of the classes of antibiotics important for human medicine, such as quinolones, are also used in animals in South East Asia [3]. In Nepal, 13% of the total veterinary expenditure was on antibiotics, the volume of which rose by 50% between 2008 and 2012 [11]. Though the use of antibiotics is rapidly increasing in food-producing animals in Nepal, there are no guidelines on use of antibiotics in food-producing animals.

One of the pillars of the WHO global action plan to tackle AMR is to optimize the use of antimicrobials in animals [1]. The Kathmandu valley of Nepal is a major hub for poultry production and consumption. Anecdotal evidence suggests widespread use of antibiotics in poultry farms in Kathmandu valley. A study from two districts of Nepal, Kailali and Kavre, showed that 39% of liver and 71% of muscle samples from broiler chickens had antibiotic residues, suggesting their widespread use [12]. A PubMed search revealed no studies assessing antibiotic use in broiler poultry farms in Nepal. Such information would be useful in informing antibiotic stewardship in animal husbandry in Nepal.

In industrial broiler poultry farms in three districts of the Kathmandu valley, we assessed the (a) prevalence of antibiotic use, (b) classes and types of antibiotics used, the purpose for their use, and (c) quantities consumed. We also inquired (d) if antibiotics were procured through prescriptions and (e) if antibiotic withdrawal periods prior to sale of flock were respected.

## 2. Materials and Methods

### 2.1. Study Design

A cross-sectional study involving primary data collection using a questionnaire through farm visits.

### 2.2. General Setting

Nepal is a landlocked country in South Asia located in the Himalayas with an estimated population of 30 million in 2020. It borders China in the north and India in the south, east, and west. Kathmandu, Pokhara, Biratnagar, and Nepalgunj are the main cities with Kathmandu being the capital. Administratively, there are seven provinces, 77 districts, and 753 municipalities in Nepal.

The agricultural sector accounts for 28% of gross domestic product (GDP), with the livestock sector contributing about 13% and poultry sector 4% of GDP [13]. Broilers are raised in almost all districts of Nepal, except for a few districts in the high Himalayas. Broiler chicken meat is a part of the staple diet in Nepal and thus available at wholesaler, retailer, and consumer outlets in all rural and urban areas.

Legislation on the use of antibiotics for animal husbandry in Nepal is governed by the Drugs Act of 1978, which covers both human and veterinary drugs. The Department of Drug Administration under the Ministry of Health and Population is authorized to implement the Drug Act. Meat and milk quality standards are regulated by the Ministry of Agriculture and Livestock Development. Quality control testing of food is done by the Department of Food Technology and Quality Control.

### 2.3. Specific Setting: Broiler Poultry Farms in the Study Districts of the Kathmandu Valley

The study sites included three districts of Kathmandu valley, namely Kathmandu, Bhaktapur, and Lalitpur. These districts are among the major broiler poultry producing sites in Nepal (Figure 1) and are also highly populated areas.

Large and small broiler poultry farms are present in Kathmandu valley. They are entirely private and owned by individuals and/or farmer cooperatives. These farms rear poultry from the stage of chicks to chicken, which takes 45 days. Subsequently, the flock are systematically sold to consumer shops. Local vendors market the poultry to slaughter houses and meat shops. Veterinarian visits are on an ad hoc basis and mostly during poultry illness or outbreaks. This can be by both government and private veterinarians and at the discretion of the farmer.

### 2.4. Study Inclusion and Period

We included all broiler farms in three districts of Kathmandu valley that had a flock size greater than 3000, referred to as “large broiler poultry farms”. These farms were included as they were registered as industrial farms and thus likely to have higher study impact. The study was conducted between July 2019 and January 2021 and field data collection was done between August and December 2019.

### 2.5. Data Collection, Sources of Data, and Statistical Analysis

A list of all broiler poultry farms in the three study districts and their flock sizes was obtained through the district veterinary offices and cross-checked through direct telephone contact with all registered broiler poultry farms. Those who met the criteria of being “large broiler poultry farms” were included in the study. The principal investigator visited all farms. The person in charge of a given farm, or the attendant was interviewed after taking informed consent. A questionnaire, which was pre-tested on three farms and adapted, was used for collecting data (see Appendix A). Data variables included: farm identification variables, flock size, antibiotics used (class and types), purpose of their use (prophylaxis and therapeutics), diseases, post-mortem, route of antibiotic administration, whether antibiotics were obtained with prescription, if antibiotic withdrawal periods were respected, and the total quantity of antibiotic for use before sale or flock culling.

Data was entered into a Microsoft Excel sheet and was imported into EpiData (version 2.2.2.183, EpiData association, Odense, Denmark) for analysis. Descriptive statistics were used to summarize the results including frequencies and proportions. Differences in means of antibiotics used by farms and between two and three districts were assessed using the t-test and ANOVA, respectively. The level of significance was set at *p* ≤ 0.05 with 95% confidence intervals.

## 3. Results

### 3.1. Prevalence, Classes, and Types of Antibiotics Used

Of 30 large broiler poultry farms included in the study, total flock size was 104,200 (range 3000–6000, mean 3473, standard deviation 906).

Table 1 shows the types of antibiotics used in the farms. A total of 27 farms used antibiotics, giving an overall prevalence of 90% (95% CI 73–98%) for antibiotic use. Six (22%) farms used antibiotics as prophylaxis, while 21 (78%) farms used them for therapeutic purposes. Seven antibiotic types from six classes were being used. The most commonly used antibiotics for treatment were tylosin (47%), colistin (47%), and dual antibiotic therapies with neomycin and doxycycline (33%). Seven (26%) farms were using more than two antibiotics during the 45 days of flock life.

None of the farms had records on the specific diseases for which prophylaxis or treatment was given. No post-mortem records of sick birds were available during the visit.

### 3.2. Quantities Consumed and Withdrawal Periods

A total of 50,000 grams of antibiotics (total weight including active and inactive antibiotic ingredients) were used for a total flock of 92,700 in 45 days of flock lifespan (0.5 grams/chicken/45 days). The Kathmandu district used the highest amount of antibiotics/chicken for 45 days. All farmers stated that they acquired antibiotics through prescriptions and were aware and respected the withdrawal period of 7–15 days prior to sale or culling. There were no records available for ascertainment of information on prescribed antibiotics and withdrawal periods.

## 4. Discussion

This is the first documented study on antibiotics use in broiler poultry farms in Nepal showing that nine in every ten large broiler poultry farms in the Kathmandu valley used antibiotics (for prophylaxis or treatment) and this included six classes and seven antibiotic types. Three of the antibiotic classes (quinolones, macrolides, and polymyxins) are considered “highest priority critically important antimicrobials” for use in humans and this has direct implications on public health [14].

The study findings are of public health importance as they indicate routine and widespread use of antibiotics in poultry farming which will inevitably lead to transmission of resistant bacteria to humans and the environment [7,8]. The most commonly used antibiotics for treatment in poultry (almost 50%) were macrolides and polymyxin. These antibiotics are last-resort treatments for multidrug resistant infections in humans and should be restricted for use in food-producing animals [5]. Macrolides (e.g., tylosin) are known to select for macrolide-resistant *Campylobacter jejuni* in poultry and at the same time are one of the few available therapies for serious campylobacter infections in children, where quinolones are not recommended [6]. Similarly, intravenous and intranasal inhalation colistin are last-resort therapies for multi-resistant *Enterobacteriaceae* and *Pseudomonas aeruginosa* infections in humans, most of which occur in intensive care units. Antibiotic resistance to colistin would render this life-saving medication ineffective for those with sepsis and pneumonia in intensive care units, including those with severe COVID-19 disease super infected with multidrug resistant bacteria [15]. The levels of tylosin and colistin use in Nepal are almost double to triple that seen in other Asian countries like the Philippines and Vietnam [16,17]. In a survey of 39 poultry farms in the Philippines, colistin use was 13%, while for tylosin this was only 8% [16]. Similarly, in a survey of 102 poultry farms in Vietnam [17], colistin use was 25.8%, while tylosin was 13.6%. While quinolone use in these two countries (3–5%) was similar to Nepal, the high levels of colistin and tylosin use in Nepal is of concern [18].

Fluoroquinolones, albeit less frequently used, are important for the treatment of urinary tract infections and essential for the construction of second line drug regimens for multidrug-resistant tuberculosis (MDR-TB) in humans [19]. This study thus highlights the importance of introducing stricter regulations, alternatives for antibiotic use, improved monitoring of its use in animal husbandry, and improving biosecurity measures in farms to prevent infections.

The study strengths are that (i) three districts and large broiler poultry farms in one of the most important poultry producing areas of Nepal were included; (ii) Kathmandu valley (the study site), is one of the most densely populated and high poultry consumption areas in Nepal with implications on One Health; (iii) we did a direct cross-verification of poultry sites with those formally registered with district authorities and all the included farms were visited by the principal investigator; and (iv) we adhered to STROBE (Structured Reporting of Observational Studies in Epidemiology) guidelines for reporting this study. Adherence to these guidelines help improve the quality of reporting research studies by ensuring transparent and completeness of reporting according to the STROBE checklist [20].

The study limitations are that since data were self-reported, we were unable to objectively ascertain if all antibiotics were procured through prescriptions, if prescriptions were justified and rational, and if withdrawal periods were being respected. We have also used total weight of antibiotics which understandably includes both active and inactive antibiotic ingredients. As our study did not use the weight of the active antibiotic agent, the antibiotic quantification should be considered with the caveat that it is likely to be exaggerated. This aspect would merit the development of standard guidelines so as to allow national and international comparisons. Finally, we are unable to demonstrate associations between the attitudes of farmers and veterinarians and use of antibiotics. This will require further research on attitudes and perceptions.

These limitations not-withstanding, the study has a number of important policy and practice implications. First, there is a need to reduce the prevalence of antibiotic use in poultry farms and urgently identify options to replace the types of antibiotics being used. WHO recommends that antibiotics used in animals should be selected from those that are “least important” to human health and not from those classified as “highest priority critically important” [5,6].

Improved hygiene, biosecurity, and better use of vaccines in poultry (e.g., against *Salmonella* species) have been shown to be effective alternatives for animal health and productivity in the absence of antibiotic use in European countries [21,22]. Best practices from these countries should be shared with Nepal and so too, peer exchange with farmers in Nepal. The One Health committee in Nepal, with technical support from tri-partite partners including the Food and Agriculture Organization (FAO), World Organization of Animal Health (OIE), and WHO, would be the platform to plan and propel this imperative, including the formulation of national guidelines for antibiotic use in poultry.

Second, there are unanswered questions relating to antibiotic use. For example, was antibiotic prophylaxis a way of masking its use as growth promoters? Were treatments being given for sporadic disease episodes or for outbreaks or to prevent disease in healthy flock? Anecdotal information suggests that fluoroquinolones and aminoglycosides are used for prophylaxis to prevent and treat *E. coli* and *Salmonella* spp. outbreaks. Tylosin is commonly used for mycoplasma related chronic respiratory disease and for necrotic enteritis associated with *Clostridium perfringens*. A combination of neomycin and doxycycline is useful for preventing and treating *Enterobacteriaceae* infections and fowl cholera. Other countries in South East Asia have reported about 84% prevalence of antibiotic use as prophylaxis and in our study, this was 22% [18]. However, our estimate might be considerably underestimated as antibiotics might already be included in poultry feeds [18], though the Department of Livestock Services has officially banned the import of antibiotics as a feed supplement. Further, we found an average of 500 mg of antibiotic used per chicken. Assuming a chicken weighed about two kilograms, this would amount to around 250 mg/kg, which is on the upper range when compared to other studies from South East Asia (range 52–276 mg) [18]. This study is unable to answer the “why” are antibiotics being used, as we do not know the exact reasons for antibiotic use. Understanding the farmer’s perspective for use of these antibiotics would be useful, and qualitative research is needed for this purpose.

Third, there is a need to get a better handle on the etiology of poultry illness. Strengthening access to and utilization of existing veterinary microbiology laboratories to understand morbidity patterns and drug resistance in poultry would help avoid unnecessary use of critically important antibiotics for humans in poultry. Improved access to veterinary professionals is also important in reaching decisions by farmers on the use of antibiotics for treatment and prevention of disease in poultry. Antibiotic awareness raising and animal health literacy promotion activities would be primordial to increase acceptance and reduce antibiotic demand among farmers. The clinical indications for the use of these antibiotics (the why?) and etiology of illness also needs further investigations.

Fourth, improved national surveillance systems would be key towards generating useful data on antibiotic use in poultry. Improved recording on common morbidities, on outbreaks, on deaths and their causes, coupled with post-mortem reports would help assess the indications for antibiotic use. A way forward would be the introduction of a monitoring register for each poultry farm which includes these and other variables to capture duration of antibiotic use in the flock as well as date of sale (or culling) to allow estimations of antibiotic withdrawal time. Data on post-mortem records should also be fed back to poultry farms as a record of the cause of disease. Such monitoring could form part of quarterly monitoring and reporting by supervising veterinarians of the Ministry of Agriculture and Livestock Development. This will also facilitate reporting by Nepal to the World Organization of Animal Health (OIE).

Finally, the information from this study indicates the need to consider updating or revising the Drug Act given almost non-existent veterinary human resources to monitor veterinary drugs [23]. Either an updated drug act with a fully functional veterinary unit with adequate veterinarians or a separate veterinary drug act may need to be formulated and implemented.

## 5. Conclusions

This study has revealed a high prevalence of multiple antibiotic use in poultry farms in the Kathmandu valley of Nepal with direct implications on One Health. The “why” behind the use of such antibiotics needs further investigation. In the meantime, steps need to be taken to address this issue and also better understand the underlying reasons behind antibiotic use in poultry farms in Nepal.

## Figures and Tables

**Figure 1 tropicalmed-06-00047-f001:**
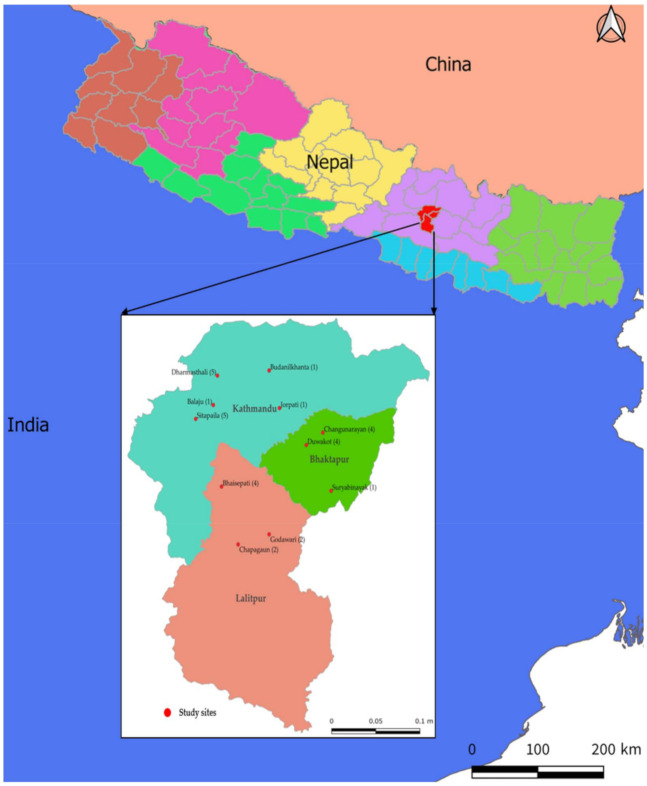
Map of Nepal showing the study sites where antibiotic use in large broiler poultry farms were assessed in Kathmandu valley, Nepal (2019).

**Table 1 tropicalmed-06-00047-t001:** Types of antibiotics used in large ^1^ broiler poultry farms in Kathmandu valley, Nepal (2019).

	Districts in Kathmandu Valley								*p*-Value ^2^
	Kathmandu		Bhaktapur		Lalitpur		Total		
	n	(%)	n	(%)	n	(%)	n	(%)	
**Total Farms**	13		9		8		30		
**Farms that used Antibiotics**	10	(77)	9	(100)	8	(100)	27	(90)	
Total Grams of Antibiotics Used ^3^	22,000		17,000		11,000		50,000		
Mean Grams per Chicken/45 Days (SD)	0.6	(0.5)	0.6	(0.2)	0.4	(0.4)	0.5	(0.3)	≤0.01
**Antibiotics for Prophylaxis**			4	(44)	2	(25)	6	(22)	
Fluoroquinolones: Enrofloxacin	-	-	1	(11)	-	-	1	(4)	
Aminoglycosides: Gentamicin	-	-	-	-	1	(13)	1	(4)	
Combination: Neomycin+Doxycycline ^4^	-	-	3	(33)	1	(13)	4	(15)	
Mean Grams per Chicken/45 Days(SD)			0.7	(0.1)	0.5	(0.2)	0.7	(0.1)	0.2
**Antibiotics for treatment ^5^**	10	(100)	5	(56)	6	(75)	21	(78)	
Macrolides: Tylosin	7	(54)	3	(33)	4	(50)	14	(47)	
Polymyxin: Colistin Sulphate	6	(46)	4	(44)	4	(50)	14	(47)	
Sulphonamides	1	(8)	4	(44)	-	-	5	(17)	
Aminoglycosides: Gentamycin	2	(15)	-	-	2	(25)	4	(13)	
Fluoroquinolones: Enrofloxacin	1	(8)	1	(11)	1	(13)	3	(10)	
Fluoroquinolones: Ciprofloxacin	1	(8)	-	-	-	-	1	(3)	
Combination: Neomycin+Doxycycline ^6^	2	(15)	5	(56)	3	(38)	10	(33)	
Mean Grams per Chicken/45 Days(SD)	0.6	(0.6)	0.4	(0.1)	0.7	(0.1)	0.5	(0.3)	0.02

SD: Standard deviation; ^1^ Flock size greater than three thousand; ^2^
*t*-test for difference in means between 2 districts; ^3^ ANOVA for difference in means between 3 districts; ^4^ Total weight including active and inactive antibiotic ingredients; ^5^ Second-line antibiotic regimen (combination of aminoglycoside and tetracyclines); ^6^ Percentages in columns may be over 100% as farms used multiple antibiotics.

## Data Availability

The data presented in this study are available on request from the corresponding author. The data are not publicly available due to restrictions and privacy.

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
