# Peer review of "Antibiotic Use in Broiler Poultry Farms in Kathmandu Valley of Nepal: Which Antibiotics and Why?"

_tropicalmed, 2021, doi:10.3390/tropicalmed6020047_

Round 1
Reviewer 1 Report
The authors did a survey of the antibiotic usage in broiler poultry farms in Nepal. This study provides valuable information for a nationwide consumption and application of antibiotics in farms and will guide the local authorities to regulate the usage of antibiotic in Nepal. The deign of experiment is well described. I would recommend publication after adding details about some terms used in the manuscript. For example, what is One Health and what is the STROBE guidelines? How does this study relate to or adhere to One Health and STROBE guidelines?
Author Response
Response to Reviewer 1
Thank you very much for reviewing this paper and your useful comments and suggestions. We have revised the manuscript in line with your suggestions. We have provided a point-by-point response to your comments and suggestions using bold font and bullets.
Response:
- One Health: as requested by the reviewer, in the Introduction (lines 44-49), we have added details on what is One Health. This makes it easier for the reader to get a good idea of what it involves
- STROBE: As requested by the reviewer, In the Discussion (Lines 209-213), we have elaborated on STROBE and explained that adherence to these guidelines help improve the quality of reporting research studies by ensuring transparent and completeness of reporting according to the STROBE checklist. We have also provided a reference 20.
- The paper does relate to One Health and we have covered this in the Discussion (lines 178-197).
Reviewer 2 Report
Dear Editor of Tropical Medicine and Infectious Disease
The manuscript title “Antibiotic use in broiler poultry farms in Kathmandu valley of Nepal: which antibiotics and why?” describes the frequency and type of antibiotic use in the intensive area of chick farms. Most chick farms used antibiotics for therapeutic indication. Tylosin and colistin were most frequently used followed by neomycin and doxycycline. The total amount of antibiotic was estimated. The study supports the gap of knowledge in antibiotic use in chick farms in the particular area. To answer why they are using in terms of type and indication, is not clarified yet.
- “Antibiotic use” includes in the Keywords.
- As the questionnaire used for farmers, it’s better to have approval from a committee on human research publication.
- Antibiotic names are a common name; the capital letter may not be proper.
- There is a lack of an association between the attitude of farmers or veterinarians to the use of antibiotics.
- It’s not clear about either t-test or ANOVA test as appropriate. The calculation seemed not to support the hypothesis.
- Nepal FDA qualifies all antibiotics used in the farms; based on bioavailability, purity, potency, etc.? This factor would be beneficial for the further analysis of the AMR rate.
- The difference in antibiotic use in each district and type of antibiotic, could be presented by a figure. The table can be used as supplementary.
- Also, the map located study area could augment the amount of antibiotic use by a shade of color.
- Many discussion texts were not directly related to the results but only adding general literature: the main output and the outcome from the monitoring data should have been concentrated. The type, indication, and frequency of antibiotic use should be compared with standard usage, nationally and internationally. It is pretty difficult to relate the result with AMR or disease outbreak when only antibiotic use data in just a local area is indicated unless the other factors associated with one-health idea include.
- According to the limitation of data availability and impact of the study. I recommended it as a Brief communication.
Author Response
Thank you very much for reviewing this paper and your useful comments and suggestions. We have revised the manuscript in line with your suggestions. We have provided a point-by-point response to your comments and suggestions using bold font and bullets.
1. Reviewer comment 1. The manuscript title “Antibiotic use in broiler poultry farms in Kathmandu valley of Nepal: which antibiotics and why?” describes the frequency and type of antibiotic use in the intensive area of chick farms. Most chick farms used antibiotics for therapeutic indication. Tylosin and colistin were most frequently used followed by neomycin and doxycycline. The total amount of antibiotic was estimated. The study supports the gap of knowledge in antibiotic use in chick farms in the particular area. To answer why they are using in terms of type and indication, is not clarified yet.
Response:
- We agree with the reviewer that this study is unable to answer the “why” are antibiotics being used as we do not know the exact reasons for antibiotic use. We have highlighted this as a limitation while suggesting further research which is already underway (lines 252-253). We have also made this clear in the conclusion (Lines 283-286)
2. Reviewer comment 2: “Antibiotic use” includes in the Keywords.
Response: As requested, we have added “Antibiotic use” in the key words
3. Reviewer comment 3. As the questionnaire used for farmers, it’s better to have approval from a committee on human research publication.
Response:
- We confirm that we have received ethics approval from a committee on human research publications. The Ethics advisory Group of the International Union Against TB and Lung Disease (EAG Approval No 66/19 dated 13/08/2019) is a human ethics research committee.
4. Reviewer comment 4. Antibiotic names are a common name; the capital letter may not be proper.
Response: We have made corrections and made all antibiotics names in small letters
5. Reviewer comment 5. There is a lack of an association between the attitude of farmers or veterinarians to the use of antibiotics.
Response:
- The study objectives did not include assessment of associations between attitudes of farmers/veterinarians and use of antibiotics. It was thus not designed for such an analysis.
- However, we have included this as an area for future research and a limitation (Lines 221-223)
6. Reviewer comment 6. It’s not clear about either t-test or ANOVA test as appropriate. The calculation seemed not to support the hypothesis.
Response: We have corrected this in the Table legend and in the methods (line 140-143). We have used t-tests to compare the differences in means between two districts and the ANOVA to compare difference in means between three districts.
7. Reviewer comment 7. Nepal FDA qualifies all antibiotics used in the farms; based on bioavailability, purity, potency, etc.? This factor would be beneficial for the further analysis of the AMR rate.
Response:
We agree, but unfortunately there are no national guidelines and such information was not available. We have thus been transparent and have highlighted this as a limitation (lines 216-221). We have also called for the development of standard guidelines in the same vein (lines 220-221)
8. Reviewer comment 8. The difference in antibiotic use in each district and type of antibiotic, could be presented by a figure. The table can be used as supplementary.
Response:
- We felt this would dilute the information in Table 1. In addition, to the quantities of antibiotics used, it is vital for the reader to visualize the types of antibiotics used as being from classes that are considered of “critical importance to human use” by WHO. Further, we have stratified the antibiotics used for prophylaxis and for treatment. As a figure, it is very complicated and less visual.
- We have thus preferred to maintain the table and thank you for your kind understanding.
9. Reviewer comment 9. Also, the map located study area could augment the amount of antibiotic use by a shade of color.
Response: This map is meant to solely highlight the study districts/Sites. It is already in color and thus we felt adding more colors would create confusion. We would thus like to maintain its purpose to the study Methods.
10. Reviewer comment 10. Many discussion texts were not directly related to the results but only adding general literature: the main output and the outcome from the monitoring data should have been concentrated. The type, indication, and frequency of antibiotic use should be compared with standard usage, nationally and internationally. It is pretty difficult to relate the result with AMR or disease outbreak when only antibiotic use data in just a local area is indicated unless the other factors associated with one-health idea include.
Response:
- Thank you for this suggestion. We have now expanded the Discussion section to include comparison of studies on antibiotics use from Philippines and Vietnam and also a review from South East Asia on antibiotic use in poultry (References 16-8, lines 191-197 and lines 245-247)
11. Reviewer comment 11. According to the limitation of data availability and impact of the study. I recommended it as a Brief communication.
Response: We have tried to keep the article as brief as possible and would be grateful to maintain it as an original article. We thank you for your kind consideration.

Round 2
Reviewer 2 Report
Dear Editor
Overall, the revision has been improved, and responded to all comments. Only the title maybe not very conclusive related to the recent finding since the article describes only the prevalence of antibiotic use in the study area. After the final revision, the editor can make your own decision without returning for the third review. I believe that this could be approved for publishing in TropicalMed.